# Rel3D: A Minimally Contrastive Benchmark for Grounding Spatial Relations in 3D

**Ankit Goyal**[†], **Kaiyu Yang**[†], **Dawei Yang**[†‡], **Jia Deng**[†]

[‡]University of Michigan, Ann Arbor, MI
[†]Princeton University, Princeton, NJ
{agoyal, kaiyuy, daweiy, jiadeng}@princeton.edu

## Abstract

Understanding spatial relations (e.g., "laptop on table") in visual input is important for both humans and robots. Existing datasets are insufficient as they lack large-scale, high-quality 3D ground truth information, which is critical for learning spatial relations. In this paper, we fill this gap by constructing Rel3D: the first large-scale, human-annotated dataset for grounding spatial relations in 3D. Rel3D enables quantifying the effectiveness of 3D information in predicting spatial relations on large-scale human data. Moreover, we propose minimally contrastive data collection—a novel crowdsourcing method for reducing dataset bias. The 3D scenes in our dataset come in minimally contrastive pairs: two scenes in a pair are almost identical, but a spatial relation holds in one and fails in the other. We empirically validate that minimally contrastive examples can diagnose issues with current relation detection models as well as lead to sample-efficient training. Code and data are available at https://github.com/princeton-vl/Rel3D.

## 1 Introduction

Spatial relations such as "laptop on table" are ubiquitous in our environment, and understanding them is vital for both humans and intelligent agents like robots. As humans, we use spatial relations for perceiving and building knowledge of the surrounding environment and supporting our daily activities such as moving around and finding objects [1, 2]. Spatial relations play an important role in communication for describing to others where objects are located [3–6].

Likewise, for robots, understanding spatial relations is necessary for navigation [7], object manipulation [8, 9], and human-robot interaction [10, 11]. For a robot to complete a task such as "put the bottle in the box", it is necessary to first understand the relation "bottle in the box".

A spatial relation is defined as a *subject-predicate-object* triplet, where *predicate* describes the spatial configuration between *subject* and *object*, such as *painting-over-bed*. Understanding spatial relations may seem an easy task at first glance, and a plausible model could be a set of hand-crafted rules for each *predicate* based on the spatial properties of *subject* and *object*, like their relative position [12–15]. However, just like many other rule-based systems, this approach works for a small set of carefully curated examples, but fails for wider real-world examples consistent with human judgment [16].

The failure results from the rich and complex semantics of spatial predicates, which depend on various factors beyond relative positions. For example, they depend on frames of reference (Is "left to the car" relative to the observer or relative to the car?), object properties (For "in front of the house", what is the frontal side of a house? Is there still a frontal side if the object were a tree?), and also commonsense ("painting over bed" is not touching while "blanket over bed" is). With all these subtleties, any set of hand-crafted rules is likely to be inadequate, so researchers have applied data-driven approaches to learn spatial relations from visual data [17, 18, 11, 7, 8, 19, 20].

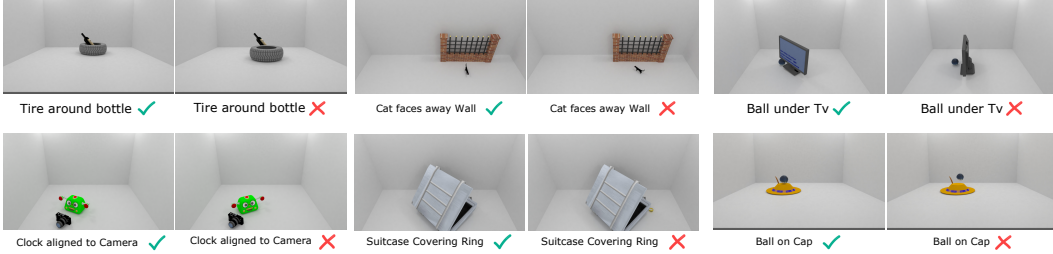

Figure 1: Some samples from Rel3D. Rel3D contains pairs of *minimally contrastive* scenes: two scenes in a pair are almost identical, but a spatial relation holds in one while fails in other.

**Benchmarking spatial relations in 3D.** Benchmark datasets have been proposed for training and evaluating a system's understanding of spatial relations [6, 19, 20]. However, existing datasets either have limited scale and variety [6, 19] or contain human-annotated relations only on 2D images [19, 20]. Prior research suggests that 3D information may play a critical role in spatial relations [20, 21, 11, 9]. With only 2D images, the model has to implicitly learn the mapping to 3D, which is itself an unsolved problem. Instead of developing an accurate 3D understanding of spatial relations, models tend to utilize superficial 2D cues [20]. More importantly, in robotics, 3D information is often readily available, making it valuable to study spatial relation recognition using 3D information.

In this work, we propose *Rel3D*—the first large-scale dataset of human-annotated spatial relations in 3D (Fig. 1). It consists of spatial relations situated in synthetic 3D scenes, making it possible to extract rich and accurate geometric and semantic information, including depth, segmentation mask, object positions, poses, and scales. The scenes in Rel3D are created by crowd workers on Amazon Mechanical Turk (Fig. 2). Workers manipulate objects according to instructions, and independent workers are asked to verify whether a given spatial relation holds in the 3D scene. We choose to use synthetic scenes because they give us complete control over various factors, e.g., objects, positions, camera poses, etc. Such flexibility is important for datasets specializing in spatial relations, enabling us to study the grounding of spatial relations with respect to these factors.

Rel3D is the first to provide rich geometric and semantic information in 3D for the task of spatial relation understanding. It enables the exploration of problems that were out of reach before. Specifically, we study how ground truth object 3D positions, scales, and poses (including frontal and upright orientation) can be used to train neural networks to predict spatial relations with high accuracy. Further, our experiments suggest that estimating 3D configurations is a promising step towards understanding spatial relations in 2D images.

**Reducing dataset bias through minimally contrastive examples.** Besides promoting 3D understanding of spatial relations, Rel3D also addresses a fundamental issue with existing datasets—biases in language and 2D spatial cues. Despite prior efforts in mitigating bias [22, 23, 20], state-of-the-art models achieve superficially high performance by exploiting bias, without much understanding [20].

We propose *minimally contrastive data collection* for spatial relations—a novel crowdsourcing method that significantly reduces dataset bias. Rel3D consists of minimally contrastive scenes: pairs of scenes with minimal differences, so that the spatial relation holds in one but fails in the other (Fig. 1). The task for the model is to classify whether the given relation holds. This minimally contrastive construction makes it unlikely for a model to exploit bias, including language bias ("cup on table" is more likely than not) and other spurious correlations with factors like the color of the background or the texture of an object. If a model attempts to associate the background with a relation, it cannot succeed in both instances of a minimally contrastive pair with identical backgrounds.

Through our experiments, we demonstrate how Rel3D can be used as an effective tool for diagnosing models that rely heavily on 2D bias as well as Language bias for making predictions. We show that a simple 2D baseline outperforms more sophisticated models, implying that these models lack 3D understanding for recognizing spatial relations. Further, we empirically demonstrate that training models on minimally contrastive examples leads to better sample efficiency.

Our contributions are as follows:

- We construct Rel3D: the first large-scale dataset of human-annotated spatial relations in 3D
- Rel3D is the first benchmark for spatial relation understanding that contains minimally contrastive examples, alleviating bias and leading to sample-efficient training

- With Rel3D, we demonstrate how 3D positions, scales, and poses of objects can be used to predict spatial relations with high accuracy

## 2 Related Work

**Spatial relations.** Research in psycholinguistics has studied how humans perceive spatial relations and use them in natural language. Landau & Jackendoff [5] have argued that natural languages use a surprisingly small set of predicates (less than 100 in English) for spatial relations, which forms the basis of how we select predicates in Rel3D. Some researchers have investigated the semantics of spatial predicates from views of human language and cognition [2–4, 16]. Others have attempted to build computational models [12, 17, 13, 15, 14]. However, these models are often based on hand-crafted rules and they are validated only on a handful of toy examples (e.g., treating objects as 2D shapes). We differ from this body of research as ours is a data-driven approach. Instead of hand-designing rules for spatial relations using a small set of curated examples, we develop machine learning models to recognize spatial relations from large-scale human-annotated visual data. We also show that our data can be potentially used to provide empirical evidence for some prior observations.

Many data-driven approaches for spatial relations have been developed in robotics, including applications in navigation [7], object manipulation [8, 18, 9] and human-robot interaction [10, 24, 11, 25, 26]. Zeng et al. [9] designed a robot to manipulate objects given visual observations of the initial state and the goal. A core intermediate step in their method is to predict spatial relations. Guadarrama et al. [11] built a robot to respond to spatial queries in natural language (e.g., "What is the object in front of the cup?"). To answer the query, the robot needs to predict spatial relations in the scene. These spatial relation modules in robotics are usually developed on small-scale datasets specific to each robotic system, making it hard to compare different methods. Also, many systems rely on hand-crafted rules that are not applicable universally [10, 24, 9]. In contrast, we build a large-scale benchmark that facilitates a common base for training and evaluating different methods.

**Visual relationship detection.** Recognizing relations in images has become a frontier of computer vision beyond object recognition. Lu et al. [27] introduced the task of visual relationship detection: the model takes an image as input and detects *subject-predicate-object* triplets by localizing pairs of objects and classifying the predicates. Since then, several datasets containing visual relations have been proposed, such as VRD [27], Visual Genome [28], and Open Images [29]. The relations in these are not necessarily spatial (e.g., "person drink tea"). We focus on spatial relations, which is an important class of visual relations; 66.0% of relations in VRD and 51.5% in Visual Genome are spatial. Unlike them, our dataset contains rich and accurate information about the 3D scene like object locations, orientation, surface normal, and depth. Further, we alleviate bias in language and 2D cues that are present in VRD and Visual Genome [20]. Many model architectures [30–38] have been developed on these datasets. We adapt and benchmark some recent works on Rel3D.

Closest to our work is SpatialSense [20], which consists of 17.5K relations on 11.6K images from NYU Depth [39] and Flickr. SpatialSense proposed adversarial crowd-sourcing to reduce language and 2D spatial bias. Rel3D differs from SpatialSense in two ways. First, Rel3D contains rich and accurate geometric and semantic information like depth, surface normal, segmentation mask, object positions, poses, and scale; while SpatialSense only contains bounding box annotations and noisy depth for some images. Rich 3d information enables analysis that is not possible with SpatialSense (see Sec.5). Second, scenes in Rel3D occur in minimally contrastive pairs. Not only does this eliminate language bias and reduce 2D bias, but it also controls for any spurious correlations with factors like background, texture, and lighting, which are not considered in SpatialSense.

**Language and 3D.** Similar to our work, prior works have also explored grounding language in 3D. Notably, Chang et al. [40] model spatial knowledge by leveraging statistics in 3D scenes. For spatial relations, they create a dataset with 609 annotations between 131 object pairs in 17 scenes. Also, Chang et al. [41] create a model for generating 3D scenes from text, and create a dataset of 1129 scenes from 60 seed sentences. Concurrent to our work, Panos et al. [42] proposed ReferIt3D, a benchmark for contrasting objects in 3D using natural and synthetic language. However, unlike in prior works [40–42], scenes in Rel3D occur in minimally contrastive pairs which control for potential biases like language bias. Also, the objects in prior works [40–42] are limited to those found in indoor scenes like chairs and tables, while Rel3D also considers outdoor objects like trees, planes, cars and birds, and hence it covers a wider array of spatial relations.

**Dataset bias.** The issue of dataset bias has plagued many machine learning tasks both within computer vision [22, 23, 20] and beyond [43–46]. Zhang et al. [22] address language bias in answering yes/no questions on clipart images. They collect pairs of images with the same question but different answers by showing the image and the question to crowd workers and asking them to modify the image so that the answer changes. Goyal et al. [23] extend this idea to real images. Unlike them, we ask for minimal modifications to input, and hence reduce bias by a much larger extent, not only in language but also in a variety of factors, including texture, color, and lighting.

"something-something" [47] is a video action recognition dataset that reduces bias by having a large number of classes. Hence, a model has to learn the action nuances (e.g., "folding something" vs. "unfolding something"). However, unlike Rel3D, it does not contain minimally contrastive pairs.

**Generating synthetic 3D data.** Our work is also related to prior works in generating synthetic 3D data using graphics engines or simulators [48–54]. This approach can produce massive data at low cost, and 3D information is readily available. It also gives us the flexibility to control various factors in the scene, such as object categories, shapes, and positions. Note that the relations in our dataset are annotated by humans rather than generated automatically.

# 3 Dataset

Rel3D consists of spatial relations situated in 3D scenes, from which one can extract rich and accurate information, such as depth, object positions, poses, and scales. Each scene contains two objects (*subject* and *object*), that either satisfy a spatial relation (*subject-predicate-object*) or not (Fig. 1). Objects in Rel3D come from multiple sources, including ShapeNet [55], and YCB [56]. We use predicates based on prior work [5] and aim to cover most of the common spatial relations. Given the vocabulary of objects and predicates, we remove triplets that are unlikely to occur in the real world (e.g. "laptop in cup"). We design an interface for crowd workers to compose 3D scenes for a given spatial relation by manipulating objects (Fig. 2). We collect instances as pairs of minimally contrastive scenes in two stages. First, we collect positive scenes wherein the spatial relation is true. Next, we give a positive scene and ask them to move the objects just enough to falsify the relation.

After collecting the 3D scenes, we render images from multiple views and conduct a final round of verification by independent crowd workers. As a result, we collect 9,990 3D scenes ( 4,995 positive, 4,995 negative) and 27336 images. The objects come from 67 categories, with 30 different spatial predicates. Below, we detail each component of our data collection pipeline.

**Object vocabulary.** The objects in our dataset are from three sources:

*ShapeNet* [55]: It is a large-scale dataset of 3D shapes. We use the ShapeNetSem subset [57], which contains rich annotations such as the frontal side, upright orientation, and real-world scale of objects. These annotations are important because, for instance, the frontal side of an object affects the configuration for the spatial predicate ``in front of'' when considered from the object's frame. There are 270 object categories in ShapeNetSem. We remove categories with too few shapes, as well as group similar ones (e.g. different types of chairs), and end up with 48 categories.

*YCB* [56]: It is a dataset for benchmarking object manipulation, consisting of everyday objects that can be manipulated on a table. These are included because object manipulation requires understanding spatial relations, and Rel3D can potentially be used for object manipulation in simulation engines. There are 77 shapes in YCB, which we manually filter and merge with ShapeNet to get 53 object categories from YCB+ShapeNet.

*Manual collection*: We collect a set of objects manually. These are from the word list of the Thing Explainer book [58]. We add 14 categories from this, including house, mountain, wall, and stick. For each category, we download five to six 3D shapes from open-source shape repositories.

In summary, the objects in Rel3D comprise of 358 shapes from 67 categories. They cover a wide range of everyday objects and are annotated with real-world scales and pose information (frontal side, upright orientation). The 3D shapes are manually reviewed to ensure quality. The train and test data contain mutually exclusive sets of 3D shapes. The supplementary material has more details.

**Predicate vocabulary.** As argued by Landau & Jackendoff [5], the space of spatial predicates is surprisingly small (less than 100 in English). We start with the list of spatial predicates in [5] and group those with similar semantic meaning (e.g., nearby and near). Next, we add multi-word prepositional phrases that describe spatial relations, such as facing towards and leaning against. We end up with 30 spatial predicates, which is a superset of those in SpatialSense [20].

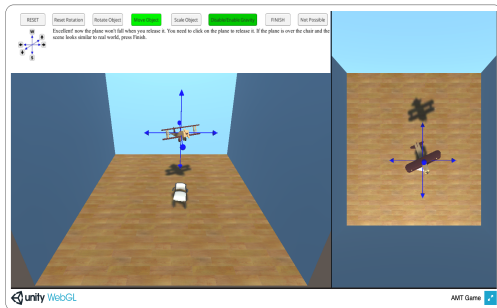

Figure 2: UI for manipulating objects to create a 3D scene given a spatial relation

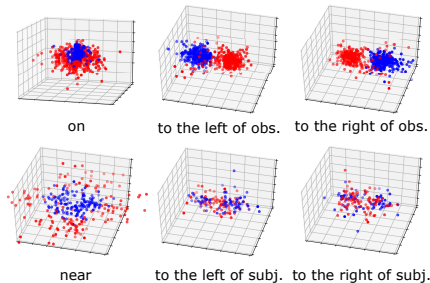

Figure 3: Each dot represent a scene in Rel3D (blue: +ve, red: -ve). The location of dot is the relative position of the object w.r.t. to the subject (subj.) in the observer's (obs.) reference frame.

Some predicates like `to the left of` are ambiguous and depend on the reference frame. For example, "person `in front of` the car" can be interpreted with respect to the observer or the car. If it is relative to the car, and the car is facing away from the observer, then the person would be behind the car in the frame of reference of the observer. Unlike prior work [20], we resolve this ambiguity by splitting such predicates into two: one relative to the *observer* and the other relative to the *object*. We ask crowd workers to complete the task without mentioning frames, and later ask which reference frame was used. This also captures real-world frequencies of frames of reference (refer to Sec. 4 for details). As a result, the spatial predicates in Rel3D are not ambiguous with respect to reference frames. However, if one wishes to retain the ambiguity, our data still captures the real-world frequencies of different reference frames.

**Relation vocabulary.** Given 67 object categories and 30 predicates, there are $67 \times 30 \times 67 = 134{,}670$ possible relations. However, not all relations are likely to happen in the real world (e.g., "laptop `in` cup"). In Rel3D, we only include relations that can occur naturally. We randomly sample about $1/4$th of all relations which then are examined by 6 expert annotators to select natural relations.

**Crowdsourcing 3D scenes.** We ask crowd workers on Amazon Mechanical Turk to compose 3D scenes by manipulating objects (Fig. 2). We create a Unity WebGL interface that renders two objects placed in an empty scene with walls and a floor. Workers can manipulate the 3D position, pose, and scale of the objects. Gravity is enabled by default but can be turned off by the worker for an object (e.g. an airplane). First, we collect positive samples. Given a spatial relation *subject-predicate-object*, the worker has to manipulate objects so that the relation holds. We ask workers to re-scale objects to resemble their real-world scales. Next, we collect minimally contrastive negative samples. Recall that a pair of minimally contrastive scenes are almost identical, but the spatial relation holds in one while fails in the other. Given a positive scene, workers are asked to move objects minimally to make the relation invalid. To simplify the task and ensure diversity, we allow them to move/rotate only one of the objects, along a randomly chosen predefined axis. If the relation cannot be invalidated by movement/rotation along the chosen axis, they can select "Not Possible", and a new axis is provided.

We find that for about 20% of negative samples, movement along the axis leads to an unnatural scene. For example, a chair could be in the air when moved along the vertical axis. While collecting negative samples, we ask AMT workers to identify these examples. Although unnatural, these are valid negative samples for the spatial relation, hence are included in the dataset. If one wishes, they can be removed using our annotations. During both the stages, we control the quality by inspecting random samples and removing annotations coming from workers with several low-quality annotations.

**Rendering and human verification.** After collecting scenes, we render images and ask independent crowd workers to verify them. For each pair of minimally contrastive scenes, we sample 12 camera views[1] and perform photo-realistic rendering using Blender [59]. The same set of camera views are used for positive and negative scenes in a pair. We show the images to crowd workers and ask them to verify whether the spatial relation holds. Each image is reviewed by three workers and we take the majority vote. We include only those image pairs for which the original labels are corroborated by the workers. This also provides us the views from which humans could distinguish whether the spatial relation holds or not. Finally, we end up with 27336 human-verified images, which we use for our

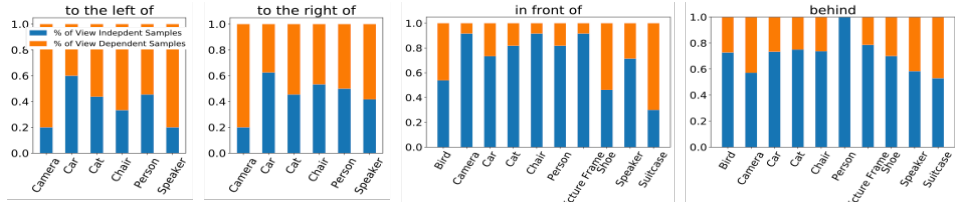

Figure 4: Percentage of intrinsic (view-independent) vs. relative (view-dependent) frames of reference for directional relations in Rel3D.

experiments. Note that, by using the 3D scene and the view information in Rel3D, one can potentially render infinitely many images by modifying factors like 3D context, background, and lighting.

**Distribution of Samples per Predicate Class.** Rel3D poses a binary classification problem where given a relation, the task is to classify whether or not the objects satisfy the relation. Rel3D has variable number of instances per predicate, as some predicates, like `on`, occur more frequently than others, like `passing through` (exact distribution in supplementary material). However, as Rel3D has each predicate represented by an equal number of positive and negative samples, the imbalance in predicate counts does not bias a model to predict one more than another. Also, Rel3D poses an independent binary classification task for each predicate and uses average class accuracy as the metric which is robust to the number of samples per predicate. Unlike Rel3D, VRD and Visual Genome use Recall@K (the recall of ground truth relations given K predicted relations), which fails to identify if a system is producing valid but unannotated predictions or false positives [20].

## 4 Dataset Analysis

**Distribution of objects in 3D space.** Since our dataset has ground truth 3D positions, it can provide insights into which regions of 3D space do humans consider as ''to the left of'' something, or how close the objects should be for them to be ''near''. In Fig. 3, we plot the relative position of the *subject* w.r.t. the *object*. Note how the directional predicate `to the left of` has different distributions depending on frames of reference. When relative to the observer, there exists a cleaner boundary of separation in the reference frame of the observer, while no such boundary exists relative to the *object*, as the object could have any orientation in the scene. Plots for other relations can be found in the supplementary material.

**Directional spatial relations.** Directional relations are spatial relations whose semantics depend on frame of reference. There are 5 directional relations in Rel3D: `to the left of`, `to the right of`, `to the side of`, `in front of`, and `behind`. They have different spatial groundings in the two frames: relative (relative to the observer) and intrinsic (relative to the *object*). Prior research in psycho-linguistics has studied the problem of how humans choose between different frames of reference [60, 12, 61, 16]. However, there aren't any empirical results based on large-scale data of human judgments. With Rel3D, we are able to shed light on this problem.

When collecting positive samples, we give the workers a relation (e.g., "person `to the left of` car") and ask them to manipulate objects to make the relation hold. We intentionally do not specify the frame of reference and let them decide. After the task, we display an image of the scene from a different viewpoint and ask if the relation remains valid. If the answer is "Yes", the worker is using intrinsic frame of reference (relative to the car). Otherwise, the worker is using relative frame of reference (relative to the observer). Based on responses from workers, our data reflects a natural distribution of how different reference frames are used. Fig. 4 shows the percentage of intrinsic reference frames (view-independent) and relative reference frames (view-dependent) for each directional relation in Rel3D. We show the plots for objects-predicate combinations with more than 10 samples for both `to the left of` and `to the right of` or both `in front of` and `behind`. We find that human choices of reference frames depend on the *object* as well as the *predicate*.

`To the left` and `to the right` have highly-correlated responses ($r = 0.79$), showing that humans make similar choices for both. The correlation between `in front of` and `behind` is 0.4, showing that they are not as symmetric as `to the left of` and `to the right of`. In fact, for some object categories like "Camera", the responses for `in front of` and `behind` are very different. This suggests that the choice of reference frames may also depend on object affordances.

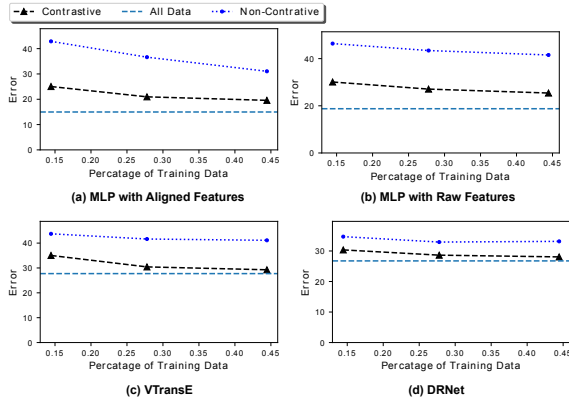

Figure 5: Contrastive (contra.) vs non contra. dataset for training. Across models, better performance is achieved with fewer samples when the dataset is contra.

Table 1: Performance on classification of spatial relations in Rel3D. Bbox means 2D bounding box and class means object class

| Model | Input | Avg. Acc. |
|---|---|---|
| Random | — | 50.00% |
| Lang Only [20] | class | 50.00% |
| BBox Only [20] | bbox | 74.14% |
| DRNet [33] | RGB + bbox + class | 73.25% |
| Vip-CNN [30] | RGB + bbox + class | 72.32% |
| VTransE [31] | RGB + bbox | 72.27% |
| PPR-FCN [34] | RGB + bbox | 73.30% |
| MLP | Raw Features | 81.24% |
| MLP | Aligned Features | 85.03% |
| Human | RGB + phrase | 94.25% |

## 5    Experiments

**Baselines for spatial relation recognition.** We benchmark state-of-the-art visual relationship detection models [33, 30, 31, 34] on Rel3D. They are outperformed by a simple baseline based solely on 2D bounding boxes, demonstrating that Rel3D is a challenging benchmark, and existing methods are unable to truly understand spatial relations.

**Experimental setup.** For benchmarking, we follow an approach similar to SpatialSense [20]. The task is spatial relation recognition: Input is an RGB image, two object bounding boxes, their category labels, and a spatial relation between them. The model predicts whether the spatial relation triplet holds in the image or not. We compute the accuracy for each of the 30 predicates separately and then report the average of those 30 values. This ensures that the reported metric reflects models' overall performance, unaffected by the variability in the number of samples per predicate.

**Model architectures.** Similar to SpatialSense, we evaluate 2D-only and language-only baselines. The 2D-only baseline takes as input the predicate and the coordinates of two bounding boxes; while the language-only baseline takes the predicate and the object categories. They output a scalar indicating whether the spatial relation holds. Similar to SpatialSense, we also adapt four state-of-the-art visual relationship detection models for our task, namely DRNet [33], Vip-CNN [30], VTransE [31] and PPR-FCN [34]. Please refer to the supplementary for more details.

**Implementation details.** All images are resized to $224 \times 224$ before feeding into the model. We perform random cropping and color jittering on training data. Hyper-parameters for each model are tuned separately using validation data, and the best-performing model on the validation set is used for testing. Please refer to the supplementary material for more details.

**Results.** Table 1 shows the performance of the baselines for spatial relation recognition on Rel3D. Accuracy for each relation can be found in the supplementary material. The dataset does not contain any language bias since each triplet (*subject-predicate-object*) has both positive and minimally contrastive negative examples. So, the language-only model does no well than a random baseline (50%). All state-of-the-art models fail to outperform the simple 2D baseline, emphasizing that the current models rely on language and 2D bias to achieve high performance on existing benchmarks. Thus, Rel3D can serve as a tool for diagnosing issues in models. Also, human performance on the dataset is around 94%. This confirms the quality of the dataset and the scope for improvement for models. The 6% errors demonstrate that some spatial relations are inherently fuzzy and subjective.

**Using 3D information for spatial relations.** A reasonable hypothesis is that the 3D configuration between objects is important in determining their spatial relation. Prior works have built computational models for spatial relations based on hand-crafted features such as angle and distance [12, 17, 13, 15, 14]. However, they are unable to provide a quantitative evaluation on large-scale natural data. Rel3D makes it possible to quantify the predictive power of 3D information for spatial relations.

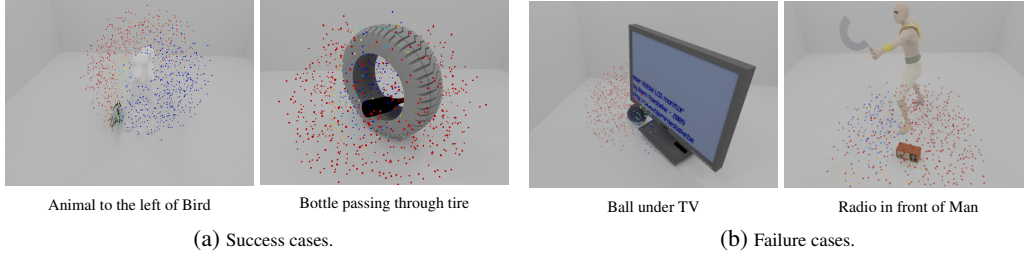

| Animal to the left of Bird | Bottle passing through tire | Ball under TV | Radio in front of Man |

(a) Success cases.　　　　　　　　　　　　　　(b) Failure cases.

Figure 6: Analyzing the success and failure cases of MLP trained with aligned features. Blue dots represent location where if object is moved, the relation would be classified as true; while red dots represent location it would be false.

To explore this, for each object in a scene, we define three reference frames: the camera reference frame $S_{\text{cam}}$, the object's raw reference frame $S_{\text{raw}}$, and the object's aligned reference frame $S_{\text{aligned}}$. In the $S_{\text{cam}}$, the camera is at origin and points towards $-z$ axis, and up direction is $y$. The object's raw reference frame $S_{\text{raw}}$ is the reference frame wherein the axes correspond to the raw CAD mesh. These axes might not be aligned for their front and up direction. If we align the $x$ axis to the mesh's frontal direction and $z$ axis to the mesh's up direction, we obtain the aligned reference frame $S_{\text{aligned}}$. We use the following two mechanisms for encoding this information:

**Raw features.** For each object, we encode its centroid in $S_{\text{cam}}$, rotation angles between $S_{\text{raw}}$ and $S_{\text{cam}}$, and scale along $xyz$ in $S_{\text{raw}}$. Since raw mesh is not aligned for the frontal and top directions, we encode this information by finding the relative rotation between $S_{\text{aligned}}$ and $S_{\text{raw}}$. The final raw feature has 24 dimensions (each object: 3-centroids, 3-rotation from $S_{\text{raw}}$ to $S_{\text{aligned}}$, 3-sizes, 3-rotation between $S_{\text{aligned}}$ and $S_{\text{raw}}$).

**Aligned features.** Here we directly encode the positions, rotation angles, and scale of the object's aligned mesh. In this way the front and up directions are implicitly encoded. We represent each object with a 9-d vector (3 - centroid in $S_{\text{cam}}$, 3 - rotation from $S_{\text{aligned}}$ to $S_{\text{cam}}$, 3 - sizes in $xyz$ directions in $S_{\text{aligned}}$). The final aligned features have 18 dimensions. Note that these features do not encode information about the exact geometry of objects. They are approximating each object as cuboids in the 3D space. We use these features to train a 5-layer Multi-layer perceptron (MLP) with skip connections for classifying spatial relations.

**Results.** The performance of different models is reported in Table 1. Further, Fig. 6a shows how our model effectively learns the decision boundary for spatial relations. Note that directly comparing these models to those using only 2D information is unfair. However, our analysis reveals how much one can gain by utilizing the 3D information. This suggests that learning to predict 3D information like pose and orientation could be an effective intermediate strategy for spatial predicate grounding.

Our results show that the 3D features alone are not sufficient to solve the relation recognition problem. In Fig. 6b, we visualize some cases where the model fails. One reason for failure is that `aligned features` do not encode information about the geometry of objects (refer to Sec. 5) but approximates each object as a cuboid. In `Ball under TV`, it approximates the TV as a cuboid and predicts some regions underneath the screen as not being under the TV. `Radio in front of Man` shows an ambiguous case where there is fuzziness whether the front of a person is defined w.r.t to their face or torso. It is important to emphasize that this study becomes possible as we have access to accurate 3D data information of the scene; and it is not possible in benchmarks that operate only in 2D images.

**Minimally Contrastive Examples Improve Sample Efficiency.** we hypothesize that minimally contrastive examples lead to sample-efficient training as they reduce bias for a network to overfit. To verify this hypothesis, we construct subsets of training data with only contrastive and only non-contrastive samples.

We construct the contrastive subset by randomly sampling minimally contrastive pairs. For the non-contrastive subset, we first sample twice as many contrastive pairs and then choose one sample from each pair. Thus the total number of training samples remains the same between contrastive and non-contrastive subsets. Fig. 5 show that training on minimally contrastive examples is much more sample efficient than on non-contrastive examples. Models trained on contrastive subsets outperform those trained on non-contrastive subsets using only about $1/4$ training data. This trend holds for models that use RGB input (DRNet and VtransE) as well as for models using 3D information. This demonstrates that minimally contrastive examples lead to sample efficient training.

# 6 Conclusion

Understanding spatial relations is an important task that requires reasoning in 3D. But existing datasets for the task lack large-scale, high-quality 3D ground truth. In this paper, we constructed Rel3D: the first large-scale dataset with human-annotated spatial relations in 3D. To reduce bias, we collected minimally contrastive pairs. Our experiments confirmed the utility of 3D information for spatial relations and the effectiveness of minimally contrastive samples for reducing bias.

# 7 Broader Impact

This work contributes to improve the understanding of spatial relations, which in turn is a critical piece of the giant puzzle on language understanding. Our work could potentially lead to better language understanding and scene comprehension for intelligent systems like robots. This can eventually help the intelligent systems to communicate better with humans. Depending on how these intelligent systems are used, the society could gain positively as well as negatively from the existence of such systems.

**Acknowledgement.** This work is partially supported by the National Science Foundation under Grant IIS-1734266 and the Office of Naval Research under Grant N00014-20-1-2634. We would like to thank Alexander Strzalkowski, Pranay Manocha and Shruti Bhargava for their help with data collection.

## Footnotes

[1]For directional relations that depend on the view of the observer, we use 3 views along the front plane.

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
