[Supplementary Material]

# Supplementary Material for Rel3D: A Minimally Contrastive Benchmark for Grounding Spatial Relations in 3D

**Ankit Goyal[†], Kaiyu Yang[†], Dawei Yang[†‡], Jia Deng[†]**
[‡]University of Michigan, Ann Arbor, MI
[†]Princeton University, Princeton, NJ
{agoyal, kaiyuy, daweiy, jiadeng}@princeton.edu

## 1 Predicate Vocabulary

There are in total 27336 images in Rel3D. Figure. 1 plots the number of images per predicate.

Figure 1: Number of images per predicate in Rel3D

## 2 Object Vocabulary

Table 1: Object categories in Rel3D along with their source.

| ShapeNetSem Only | YCB Only | Both in YCB and ShapeNetSem | SimpleWord |
|---|---|---|---|
| Speaker | Can | Cereal Box | Building |
| Computer | Fruits | Bottle | Mountain |
| Picture Frame | Cup | | Glasses |
| Desk | Block | | Radio |
| TV | Ball | | Stick |
| Mirror | | | Stone |
| Person | | | House |
| Picture | | | Wheel |
| Book | | | Shoe |
| Chair | | | Door |
| Rug | | | Tire |
| Bed | | | Tree |
| Camera | | | Wall |
| Media Storage | | | Bag |
| Table | | | |
| Plant | | | |
| Gun | | | |
| CellPhone | | | |
| Plate | | | |
| Cat | | | |
| Controller | | | |
| Knife | | | |
| Cap | | | |
| Animal | | | |
| Clock | | | |
| Ladder | | | |
| Car | | | |
| Washer | | | |
| Bus | | | |
| Boat | | | |
| Fish | | | |
| Ring | | | |
| Truck | | | |
| Bird | | | |
| Teapot | | | |
| Airplane | | | |
| Bowl | | | |
| Fork | | | |
| Spoon | | | |
| Couch | | | |
| Child Bed | | | |
| Vase | | | |
| Toilet | | | |
| Sink | | | |
| Suitcase | | | |
| Bike | | | |

Table 2: Mapping between categories in ShapeNetSem and Rel3D. All categories in ShapeNetSem not present in Rel3D either have few shapes or have spatial relations similar to some category in Rel3D.

| ShapeNetSem Cat. | In Rel3D | < 5 shapes | Rel3D Cat. | Similar Cat. in Rel3D |
|---|---|---|---|---|
| WallArt | Yes | | Picture | |
| Chair,Recliner | Yes | | Chair | |
| Speaker | Yes | | Speaker | |
| Lamp,DeskLamp | No | | | Plant |
| Chair,OfficeChair | Yes | | Chair | |
| Chair,rankChairs,OfficeSideChair | Yes | | Chair | |
| Computer | Yes | | Computer | |
| Chair,rankChairs | Yes | | Chair | |
| Table,RoundTable,AccentTable | Yes | | Table | |
| Picture Frame | Yes | | Picture Frame | |
| Couch | Yes | | Couch | |
| Candle | No | | | Bottle |
| Dresser,ChestOfDrawers | Yes | | Media Storage | |
| Desk | Yes | | Desk | |
| Helicopter | Yes | | Airplane | |
| Bottle,WineBottle | Yes | | Bottle | |
| Clock,StandingClock | Yes | | Clock | |
| Bench | Yes | | Table | |
| TV | Yes | | TV | |
| Mirror | Yes | | Mirror | |
| Person | Yes | | Person | |
| Picture | Yes | | Picture | |
| Bottle | Yes | | Bottle | |
| Shelves,Bookcase | Yes | | Media Storage | |
| TrashBin | No | | | Cup |
| Bed,LoftBed | No | | Bed | |
| Books | Yes | | Book | |
| Chair,OfficeSideChair | Yes | | Chair | |
| Book | Yes | | Book | |
| Chair | Yes | | Chair | |
| Lamp,FloorLamp | No | | | Plant |
| Laptop,Computer | Yes | | Computer | |
| FoodItem,MilkCarton | Yes | | Cereal Box | |
| Oven,Counter | No | Yes | | Washer |
| Fireplace | No | | | |
| Lamp,TableLamp | No | | | Plant |
| Rug | Yes | | Rug | |
| ArcadeMachine | No | Yes | | |
| Monitor | Yes | | TV | |
| LightSwitch | No | | | Block |
| OutdoorTable | Yes | | Table | |
| Bed | Yes | | Bed | |
| Chair,rankChairs,OfficeChair | Yes | | Chair | |
| Table,AccentTable | Yes | | Table | |
| Camera | Yes | | Camera | |
| Media Storage | Yes | | Media Storage | |
| Table | Yes | | Table | |
| Plant | Yes | | Plant | |
| Gun | Yes | | Gun | |
| Dresser,ChestOfDrawers,DresserWithMirror | Yes | | Media Storage | |
| Child Bed | Yes | | Child Bed | |
| Cabinet | Yes | | Media Storage | |
| Vase | Yes | | Vase | |
| Cabinet,Sideboard | Yes | | Media Storage | |
| Chair,Recliner,rankChairs | Yes | | Chair | |

| | | | | |
|---|---|---|---|---|
| Lamp,WallLamp | No | | | Plant |
| TvStand,Media Storage | Yes | | Media Storage | |
| Window | No | | | Picture |
| Stapler | No | | | Block |
| Microwave | No | | | Washer |
| Violin | No | | | Knife |
| TvStand,MediaStorage | Yes | | Media Storage | |
| Table,DiningTable | Yes | | Table | |
| Table,RoundTable | Yes | | Table | |
| CellPhone | Yes | | CellPhone | |
| Chair,SideChair | Yes | | Chair | |
| Plate | Yes | | Plate | |
| Teapot | Yes | | Teapot | |
| Cat | Yes | | Cat | |
| Lamp,CeilingLamp | No | | | Plant |
| Controller | Yes | | Controller | |
| Armoire,ChestOfDrawers | Yes | | Media Storage | |
| Chair,ChairWithOttoman | Yes | | Chair | |
| Lamp,LampPost | No | | | Plant |
| Armoire,Dresser,Wardrobe,ChestOfDrawers | Yes | | Media Storage | |
| Armoire,Wardrobe,ChestOfDrawers | Yes | | Media Storage | |
| Picture Frame | Yes | | Picture Frame | |
| Bookcase | Yes | | Media Storage | |
| Oven | No | | | Washer |
| Refrigerator | No | | | Washer |
| Table,EndTable | Yes | | Table | |
| Table,RoundTable,DiningTable | Yes | | Table | |
| Lamp,DeskLamp,FloorLamp | No | | | Plant |
| Piano | No | | | Table |
| Bathtub | No | Yes | | |
| Stool | Yes | | Table | |
| Whiteboard | No | | Picture | |
| Knife | Yes | | Knife | |
| Lamp | No | | | Plant |
| Donut,FoodItem | No | | | Ring |
| Cap | Yes | | Cap | |
| GameTable | Yes | | Table | |
| Chair,Couch | Yes | | Chair | |
| ChestOfDrawers | Yes | | Media Storage | |
| Table,EndTable,RoundTable | Yes | | Table | |
| Table,CoffeeTable | Yes | | Table | |
| MediaPlayer | No | | | Washer |
| MousePad | No | Yes | | |
| Animal | Yes | | Animal | |
| Picture,Painting | Yes | | Picture | |
| Airplane | Yes | | Airplane | |
| Toilet | Yes | | Toilet | |
| Chair,AccentChair | Yes | | Chair | |
| FoodItem | No | Yes | | |
| Computer | Yes | | Computer | |
| Printer | No | | | Washer |
| DartBoard | No | | | WallArt |
| Curtain | No | | | Door |
| Booth | No | Yes | | |
| SpaceShip | Yes | | Airplane | |
| Horse | Yes | | Animal | |

| | | | | |
|---|---|---|---|---|
| Nightstand,ChestOfDrawers | Yes | | Media Storage | |
| Tank | No | | | Truck |
| FileCabinet | Yes | | Media Storage | |
| Bed,PosterBed | Yes | | Bed | |
| Cabinet,CurioCabinet | Yes | | Media Storage | |
| Table,RoundTable,CoffeeTable | Yes | | Table | |
| Sink | Yes | | Sink | |
| Ipod | Yes | | CellPhone | |
| Bed,SingleBed | Yes | | Bed | |
| Animal,PlushToy | Yes | | Animal | |
| ToyFigure | No | | | Block, Person |
| Bed,KingBed | Yes | | Bed | |
| TvStand | No | Yes | | |
| Notepad | Yes | | Book | |
| Dryer | Yes | | Washer | |
| WallArt,Painting | Yes | | Picture | |
| Bottle,BeerBottle | Yes | | Bottle | |
| NintendoDS,VideoGameController | Yes | | Controller | |
| BarTable | Yes | | Table | |
| Bowl | Yes | | Bowl | |
| Chair,Chaise,rankChairs | Yes | | Chair | |
| Clock | Yes | | Clock | |
| Stool,Barstool | Yes | | Table | |
| Toy | No | Yes | | |
| Clock,WallClock | Yes | | Clock | |
| Rock | Yes | | Stone | |
| Bottle,DrinkBottle | Yes | | Bottle | |
| Dart | No | Yes | | |
| Fork | Yes | | Fork | |
| Cabinet,FileCabinet | No | | Media Storage | |
| CoffeeMaker | No | | | Sink |
| DresserWithMirror | Yes | | Mirror | |
| Ladder | Yes | | Ladder | |
| Chair,Chaise | Yes | | Chair | |
| Poster,WallArt | Yes | | Picture | |
| Couch,Loveseat | Yes | | Couch | |
| Chair,BeanBag | Yes | | Chair | |
| Suitcase | Yes | | Suitcase | |
| Wii,VideoGameConsole | Yes | | Controller | |
| Dog | Yes | | Animal | |
| PianoKeyboard | No | | | Controller, Block |
| PSP,VideoGameController | Yes | | Controller | |
| Snowman | No | Yes | | |
| Cabinet,Sideboard,CurioCabinet | No | Yes | | |
| Dishwasher | Yes | | Washer | |
| Spoon | Yes | | Spoon | |
| Horse,ToyFigure | Yes | | Animal | |
| Radio | Yes | | Radio | |
| Car | Yes | | Car | |
| FoodItem,Cereal Box | Yes | | Cereal Box | |
| Painting | Yes | | Picture | |
| Washer | Yes | | Washer | |
| Bus | Yes | | Bus | |
| Bed,DoubleBed | Yes | | Bed | |
| Cereal Box | Yes | | Cereal Box | |
| ComputerMouse | Yes | | | |

| | | | | |
|---|---|---|---|---|
| ToyFigure,SpaceShip | No | | | |
| Bowl,FoodItem | No | | Bowl | |
| SodaCan | Yes | | Can | |
| Boat | Yes | | Boat | |
| Fish | Yes | | Fish | |
| GameTable,RoundTable | Yes | | Table | |
| Coin | No | | | Plate, Ring |
| Picture,WallArt,Painting | Yes | | Picture | |
| Airplane,ToyFigure,rankAirplanes | Yes | | Airplane | |
| MediaDiscs | No | Yes | | |
| Ipad | Yes | | CellPhone | |
| Ring | Yes | | Ring | |
| PowerSocket | No | Yes | | |
| Ipod,Cabling | No | Yes | | |
| _StanfordSceneDBModels | No | | | |
| Couch,Sectional | Yes | | Couch | |
| Backpack | Yes | | Bag | |
| Truck | Yes | | Truck | |
| Lamp,DeskLamp,TableLamp | No | Yes | | |
| Elephant | Yes | | Animal | |
| Bed | Yes | | Bed | |
| Desk,DraftingTable | Yes | | Desk | |
| Bed,RoundBed | Yes | | Bed | |
| Pen | No | Yes | | |
| Chair,SideChair,rankChairs | Yes | | Chair | |
| Board | Yes | | Picture | |
| Bowl,FruitBowl | Yes | | Bowl | |
| Keyboard | No | | | Block, Book |
| Door | Yes | | Door | |
| Chair,rankChairs,AccentChair | Yes | | Chair | |
| ChestOfDrawers,DresserWithMirror | Yes | | Media Storage | |
| Ipod,Headphones | No | Yes | | |
| DiscCase | No | Yes | | |
| Tank,ToyFigure | No | Yes | | |
| Headphones | Yes | | | |
| LDesk | No | | Desk | |
| Counter | No | | Table | |
| Nightstand | No | Yes | | |
| Rack,Shelves,CoatRack | Yes | | Media Storage | |
| Sword | Yes | | Knife | |
| Ship | Yes | | Boat | |
| Hanger | No | Yes | | |
| Easel | No | Yes | | |
| Bed,CanopyBed,PosterBed | No | | Bed | |
| Armoire,Wardrobe | No | Yes | | |
| Vanity | No | Yes | | |
| SkateBoard | No | Yes | | |
| Bird | Yes | | Bird | |
| BedWithNightstand | Yes | | Bed | |
| Toy,Clock | Yes | | Clock | |
| Car,ToyFigure | Yes | | Car | |
| Cow | Yes | | Animal | |
| Shelves | Yes | | Media Storage | |
| Camera,DSLRCamera | Yes | | Camera | |
| Bike | Yes | | Bike | |
| Bicycle | Yes | | Bike | |

| | | | | |
|---|---|---|---|---|
| Thumbtack | No | Yes | | |
| Picture,WallArt | No | | Picture | |
| Shirt | No | Yes | | |
| _OIMwhitelist | No | Yes | | |
| Rack | No | Yes | | |
| Fries,FoodItem | No | Yes | | |
| Hat | Yes | | Cap | |
| Bowl,FoodItem,FruitBowl | Yes | Yes | Bowl | |
| Armoire | Yes | | Media Storage | |
| PowerStrip | No | Yes | | |
| Chair,KneelingChair | Yes | | Chair | |
| Harp | No | Yes | | |
| Rack,Shelves | Yes | | Media Storage | |
| Armoire,Dresser,ChestOfDrawers | Yes | | Media Storage | |
| Table,ChestOfDrawers | Yes | | Table | |
| Toy,Horse | Yes | Yes | Animal | |
| Knife,Utensils | Yes | | Knife | |
| Ipod,CellPhone | Yes | | CellPhone | |
| Gamecube,VideoGameConsole | Yes | | Controller | |
| StaplerWithStaples | No | Yes | | |
| Cabinet,ChestOfDrawers | Yes | | Media Storage | |
| Toy,Truck | Yes | | Truck | |
| Bowl,Plate | Yes | | Bowl | |
| Faucet | Yes | Yes | Sink | |
| Kettle | Yes | | Teapot | |
| Toy,Animal,PlushToy | Yes | | Animal | |
| VideoGameConsole | Yes | | Controller | |
| Bed,Crib | Yes | | Child Bed | |
| ToiletPaper | No | | | |
| WasherDryerSet | Yes | | Washer | |
| Donkey | Yes | | Animal | |
| TissueBox | No | Yes | | |
| RiceCooker | No | Yes | | |
| Credenza | No | Yes | | |
| Bed,KingBed | Yes | | Bed | |
| Plate,FoodItem,FoodPlate | Yes | | Plate | |
| Poster | Yes | | Picture | |
| Shower | No | Yes | | |
| Lectern | No | Yes | | |
| Cassette | No | Yes | | |
| WallArtWithFigurine | Yes | | Picture | |
| Glasses | Yes | | Glasses | |
| TvStand,RoundTable,MediaStorage | Yes | | Table | |
| Xbox,VideoGameConsole | No | Yes | | |
| GuitarStand | No | Yes | | |
| _BAD | No | Yes | | |
| Telephone | No | Yes | | |
| Bed,Cradle | Yes | | Child Bed | |
| Toy,Elephant,PlushToy,ToyFigure | Yes | | Animal | |
| Bidet | No | Yes | | |
| DeskLamp | No | | | Plant |
| TV,TvStand | No | Yes | | |
| Radio,Clock | Yes | | Clock | |
| Bed,Dresser,DoubleBed,ChestOfDrawers | Yes | | Bed | |
| Room | No | | | |
| Calculator | No | | | Block, Controller |

| Toy,Airplane | | No | | Airplane | |
|---|---|---|---|---|---|

Table 3: Mapping between objects in YCB and Rel3D.

| Object ID | Rel3D Category | Reason for not including in Rel3D |
|---|---|---|
| 001_chips_can | Can | |
| 002_master_chef_can | Can | |
| 003_cracker_box | Cereal Box | |
| 004_sugar_box | Cereal Box | |
| 005_tomato_soup_can | Can | |
| 006_mustard_bottle | Bottle | |
| 007_tuna_fish_can | Can | |
| 008_pudding_box | Cereal Box | |
| 009_gelatin_box | Cereal Box | |
| 010_potted_meat_can | Can | |
| 011_banana | Fruits | |
| 012_strawberry | Fruits | |
| 013_apple | Fruits | |
| 014_lemon | Fruits | |
| 015_peach | | Bad reconstruction |
| 016_pear | Fruits | |
| 017_orange | Fruits | |
| 018_plum | | Bad reconstuction |
| 019_pitcher_base | | Bad reconstuction |
| 021_bleach_cleanser | Bottle | |
| 022_windex_bottle | | Largely distorted |
| 023_wine_glass | | Missing Object |
| 024_bowl | | Bad reconstruction |
| 025_mug | Cup | |
| 026_sponge | Block | |
| 027_skillet | | Bad reconstuction |
| 028_skillet_lid | | Missing object |
| 029_plate | | Bad reconstuction |
| 030_fork | | Bad reconstuction |
| 031_spoon | | Bad reconstuction |
| 032_knife | | Bad reconstuction |
| 033_spatula | | Missing object |
| 035_power_drill | | Only 1 shape |
| 036_wood_block | Block | |
| 037_scissors | | Only 1 shape |
| 038_padlock | | Bad reconstruction |
| 039_key | | Missing object |
| 040_large_marker | | Only 1 shape |
| 041_small_marker | | Bad reconstuction |
| 042_adjustable_wrench | | Bad reconstuction |
| 043_phillips_screwdriver | | Bad reconstuction |
| 044_flat_screwdriver | | Bad reconstuction |
| 048_hammer | | Only 1 shape |
| 049_small_clamp | | Missing .obj file |
| 050_medium_clamp | | Bad reconstuction |
| 051_large_clamp | | Bad reconstuction |
| 052_extra_large_clamp | | Only 1 Shape |
| 053_mini_soccer_ball | | Bad reconstuction |
| 054_softball | Ball | |
| 055_baseball | Ball | |
| 056_tennis_ball | Ball | |

| | | |
|---|---|---|
| 057_racquetball | Ball | |
| 058_golf_ball | Ball | |
| 059_chain | | Bad reconstruction |
| 061_foam_brick | Block | |
| 062_dice | | Bad reconstruction |
| 063-a_marbles | | Bad reconstruction |
| 063-b_marbles | | Missing objects |
| 063-c_marbles | | Missing objects |
| 063-d_marbles | | Missing objects |
| 063-e_marbles | | Missing objects |
| 063-f_marbles | | Missing objects |
| 065-a_cups | | Bad reconstruction, looks like a blob |
| 065-b_cups | | Bad reconstruction |
| 065-c_cups | | Bad reconstruction |
| 065-d_cups | Cup | |
| 065-e_cups | Cup | |
| 065-f_cups | Cup | |
| 065-g_cups | Cup | |
| 065-h_cups | | Bad reconstruction |
| 065-i_cups | Cup | |
| 065-j_cups | Cup | |
| 070-a_colored_wood_blocks | | Bad reconstruction |
| 070-b_colored_wood_blocks | | Missing objects |
| 071_nine_hole_peg_test | Block | |
| 072-a_toy_airplane | | Small Object Part |
| 072-b_toy_airplane | | Small Object Part |
| 072-c_toy_airplane | | Small Object Part |
| 072-d_toy_airplane | | Small Object Part |
| 072-e_toy_airplane | | Small Object Part |
| 072-f_toy_airplane | | Small Object Part |
| 072-g_toy_airplane | | Small Object Part |
| 072-h_toy_airplane | | Small Object Part |
| 072-i_toy_airplane | | Small Object Part |
| 072-j_toy_airplane | | Small Object Part |
| 072-k_toy_airplane | | Small Object Part |
| 073-a_lego_duplo | | Small Object Part |
| 073-b_lego_duplo | | Small Object Part |
| 073-c_lego_duplo | | Small Object Part |
| 073-d_lego_duplo | | Small Object Part |
| 073-e_lego_duplo | | Small Object Part |
| 073-f_lego_duplo | | Small Object Part |
| 073-g_lego_duplo | | Small Object Part |
| 073-h_lego_duplo | | Small Object Part |
| 073-i_lego_duplo | | Small Object Part |
| 073-j_lego_duplo | | Small Object Part |
| 073-k_lego_duplo | | Small Object Part |
| 073-l_lego_duplo | | Small Object Part |
| 073-m_lego_duplo | | Small Object Part |
| 076_timer | | Only 1 shape |
| 077_rubiks_cube | Block | |
| 078_tshirt | | No obj model |

# 3 Relation Plots

(a) aligned to

(b) around

(c) behind (wrt you)

(d) behind

(e) covering

(f) faces away

(g) faces towards

(h) far from

(i) in front of (wrt you)

(j) in front of

(k) in

(l) inside

(m) leaning against

(n) near

(o) on top of

(p) on

(q) outside

(r) over

(s) passing through

(t) points away

(u) points towards

(v) to the left of (wrt you)

(w) to the left of

(x) to the right of (wrt you)

(y) to the right of

(z) to the side of (wrt you)

(aa) to the side of

(ab) touching

(ac) under

Figure 2: Each dot represents a scene in our dataset (blue means positive examples and red for negative). The location of the dot represent the relative position of the object w.r.t. to the subject in the frame of reference of the observer.

## 4 Model Hyperparameters

Table 4: All the hyper-parameters we used to tune the baseline models in Rel3D. The values in bold represent the hyperparameter choice that performed the best on the validation set.

| Model | l2 regularization | Feature dim | Roi size | Back-bone |
|---|---|---|---|---|
| 2D | 0, **1e-4**, 1e-3 | 64, 128, **256**, 512 | | |
| VtranE | 0, 1e-6, **1e-4**, 1e-2 | 128, 256, 512, 1024, **2048**, 4096 | 1, **3**, 5 | **resnet18** |
| VipCNN | **0**, 1e-6, 1e-4 | | 3, 5, **7**, 9 | **resnet18**, resnet101 |
| DRNet | 0, 1e-6, **1e-4**, 1e-3 | 64, **128**, 256, 512 | | |
| PPFRCN | **0**, 1e-6, 1e-4 | | **3** | **resnet18**, resnet101 |

## 5 Results Predicate-wise

| Model | aligned to | around | behind | behind (wrt you) | below | covering | faces away | faces towards | far from | in | in front of | in front of (wrt you) | inside | leaning against | near | on | on top of | outside | over | passing through | points away | points towards | to the left of | to the left of (wrt you) | to the right of | to the right of (wrt you) | to the side of | to the side of (wrt you) | touching |
|---|---|---|---|---|---|---|---|---|---|---|---|---|---|---|---|---|---|---|---|---|---|---|---|---|---|---|---|---|---|
| 2d | 0.578 | 0.817 | 0.576 | 0.945 | 0.845 | 0.873 | 0.513 | 0.5 | 0.872 | 0.762 | 0.625 | 0.931 | 0.717 | 0.757 | 0.916 | 0.845 | 0.831 | 0.667 | 0.798 | 0.67 | 0.516 | 0.551 | 0.717 | 0.948 | 0.517 | 0.935 | 0.661 | 0.889 | 0.674 |
| VtranE | 0.612 | 0.768 | 0.53 | 0.961 | 0.818 | 0.822 | 0.526 | 0.55 | 0.723 | 0.766 | 0.578 | 0.944 | 0.692 | 0.716 | 0.878 | 0.849 | 0.838 | 0.667 | 0.805 | 0.717 | 0.548 | 0.543 | 0.667 | 0.933 | 0.542 | 0.952 | 0.641 | 0.792 | 0.667 |
| VipCNN | 0.647 | 0.774 | 0.606 | 0.914 | 0.75 | 0.803 | 0.545 | 0.567 | 0.872 | 0.703 | 0.602 | 0.889 | 0.725 | 0.662 | 0.882 | 0.797 | 0.77 | 0.583 | 0.743 | 0.594 | 0.492 | 0.623 | 0.5 | 0.881 | 0.592 | 0.917 | 0.713 | 0.889 | 0.727 |
| DRNet | 0.612 | 0.848 | 0.576 | 0.961 | 0.777 | 0.866 | 0.539 | 0.558 | 0.723 | 0.752 | 0.625 | 0.958 | 0.75 | 0.676 | 0.849 | 0.832 | 0.858 | 0.708 | 0.775 | 0.689 | 0.5 | 0.529 | 0.6 | 0.952 | 0.475 | 0.957 | 0.67 | 0.875 | 0.667 |
| PPFRCN | 0.647 | 0.811 | 0.606 | 0.844 | 0.777 | 0.793 | 0.513 | 0.567 | 0.83 | 0.752 | 0.633 | 0.889 | 0.7 | 0.635 | 0.857 | 0.828 | 0.791 | 0.542 | 0.741 | 0.66 | 0.516 | 0.601 | 0.817 | 0.913 | 0.558 | 0.939 | 0.69 | 0.903 | 0.667 |
| MLP (Aligned Absolute) | 0.672 | 0.945 | 0.803 | 0.977 | 0.946 | 0.904 | 0.955 | 0.783 | 0.787 | 0.841 | 0.891 | 0.944 | 0.817 | 0.797 | 0.929 | 0.909 | 0.824 | 0.792 | 0.869 | 0.698 | 0.913 | 0.703 | 0.817 | 0.956 | 0.808 | 0.965 | 0.796 | 0.889 | 0.674 |
| MLP (Raw Absolute) | 0.672 | 0.915 | 0.682 | 0.977 | 0.946 | 0.889 | 0.818 | 0.608 | 0.862 | 0.831 | 0.742 | 0.903 | 0.85 | 0.743 | 0.933 | 0.888 | 0.845 | 0.833 | 0.874 | 0.708 | 0.786 | 0.652 | 0.717 | 0.94 | 0.567 | 0.987 | 0.716 | 0.917 | 0.659 |
| Human | 0.888 | 0.982 | 0.97 | 0.961 | 0.946 | 0.965 | 0.968 | 0.925 | 0.777 | 0.955 | 0.938 | 0.958 | 0.933 | 0.932 | 0.966 | 0.983 | 0.973 | 0.75 | 0.982 | 0.972 | 0.9682 | 0.986 | 0.95 | 0.988 | 0.958 | 0.983 | 0.951 | 0.986 | 0.932 |