[Reviews · NeurIPS 2020]

Review 1

Summary and Contributions: Te paper presents a new open-source dateset, Rel3D, of synthetic 3D scenes annotated with pairwise spatial object relationships. Scene in the sets come in contrastive pairs that are minimally different, but where the relationship holds in one member and not in the other of the pair. Such a data set can be used for grounding language that refers to to spatial object relationships and lead to improved visual relationship detection. The work specifically aims to address shortcomings of current data sets that are only in 2D and where data set bias is a strong contributor to performance.

Strengths: The authors have produced a modestly large 3D scene data set (about 10K scenes) in pairs of positive and negative relationships. The authors thus have taken care to generate a data set that gives as much weight to negative examples as to positive ones. They have also dealt with various language ambiguity issues, as spatial relationships for a given view may be based either on the observer's frame or the object's frame of reference. The authors argue, and demonstrate by a small study, the advantage of 3D data for determining spatial relationships over purely 2D approaches. They also show that their minimally contrastive examples allow learning with increased sample efficiency. After reading the other reviews and the authors' detailed and thoughtful and detailed rebuttal I remain confident in my positive assessment. The Ref3D object relations data set is a well carried out data collection effort that should be very useful to those working at the interface of 3D (and 2D) computer vision with grounded language. The balanced contrastive pairs make it especially appealing for efficient learning. I hope that in the future the authors will expand the data set to more realistic scenes with multiple objects.

Weaknesses: The author's scenes involve only two objects each and are thus not so realistic.

Correctness: Their methodology for data collection is well explained and founded.

Clarity: The paper is very well written.

Relation to Prior Work: Yes.

Reproducibility: Yes

Additional Feedback:


Review 2

Summary and Contributions: This paper provides a new dataset that is annotated with intensive relation labels and designs a corresponding benchmark to support spatial relation reasoning. This dataset takes into account various relation scenarios with comprehensive 'subject-predicate-object' triplets. Since it is built in a synthetic manner, it provides rich 3D spatial clues (depth, surface normals, 3D configurations, etc.) beyond 2D images, that support relation reasoning with different modalities. Although it is constructed with synthetic models, the relation labels are annotated with human labors rather than using a set of hand-designing rules and takes into account human subjective factors into experiments. To mitigate data imbalance, the dataset is designed with a minimally contrastive approach that leads to sample-efficient training.

Strengths: This paper provides a 3D relation dataset with comprehensive human annotations. This dataset is built with a large scale of shape models (subjective and objectives) and relation predicates. It is also well-tailored with minimally contrastive design to mitigate data bias. The 3D information also supports relation inference using geometric data to predict spatial relations with high accuracy.

Weaknesses: The main contribution of this paper is the well tailored dataset Rel3D. For the methods in the benchmark (Table 1), it validates that many 2D methods rely on the language and 2D bias to achieve a high accuracy on existing datasets but fails on Rel3D. It also indicates leveraging 3D features can largely improves the relation inference. However, this paper does not provide a proper solution to fully leverage this dataset to advance the relation reasoning from 2D.

Correctness: Claims and methods in this paper are clarified clearly.

Clarity: The paper is well written and easy to follow.

Relation to Prior Work: Yes

Reproducibility: Yes

Additional Feedback: Post-rebuttal: After reading the rebuttal and others' comments, I would like to raise my rating. I am convinced that Rel3D will be helpful to trigger researchs in the relevant area.


Review 3

Summary and Contributions: The paper presents a dataset of synthetic 3D object pairs exhibiting a variety of spatial relations. The dataset is constructed by crowdsourcing the placements of object pairs given a particular subject-predicate-object spatial relation. Crowdworkers are then presented with rendered images of the object pairs to verify whether the relation holds. Moreover, the crowdworkers create minimal contrastive pairs by perturbing an object pair to falsify the specified spatial relation. A number of recent methods are applied on the dataset to benchmark their performance on the spatial relation classification task (binary yes/no output). It is shown that recent methods are outperform by a naive baseline using only 2D bounding box information. The authors propose an approach that leverages 3D information (e.g., object relative location and pose), which improves performance but still does not reach human-level spatial relation classification performance. Other experiment shows that the minimally contrastive pairs lead to improved sample efficiency.

Strengths: + Dataset collection procedure is described comprehensively, with clear rationales for most design decisions + The construction of minimally contrastive pairs is clever and the resulting data appears to be highly valuable

Weaknesses: - There needs to be a discussion of how this work relates to earlier work by Chang et al. on crowdsourcing synthetic 3D scenes to investigate language grounding (including spatial relation grounding) -- see the detailed comments on related work. - A few details are missing from the exposition and should be clarified: 1) how were specific object instances selected and were they pre-specified for a given crowdworker? 2) Rationale for subsampling only 1/4 of all relations and what relations are excluded due to that? 3) To my understanding, a crowdworker's judgment that a relation is not valid in a presented image is deemed to imply that the relation is not using an intrinsic frame of reference -- is this a reasonable assumption (i.e. are there other cases when a crowdworker might say the relation does not hold)?

Correctness: Yes, as far as I can tell.

Clarity: Yes, the paper is well written.

Relation to Prior Work: Yes, mostly. There are a couple missing references to prior work that has addressed constructed synthetic 3D scene datasets and/or addressed 3D spatial relations: - Learning Spatial Knowledge for Text to 3D Scene Generation [Chang et al. EMNLP 2014] - Text to 3D Scene Generation with Rich Lexical Grounding [Chang et al. ACL 2015]

Reproducibility: Yes

Additional Feedback: Some minor writing issues: 72: "Language" -> language 169: "comprise of 358 shapes" 286: "does no well than a random baseline" 329: "Sample Efficiency. we hypothesize" -> We hypothesize *** Post-rebuttal update I have read the rebuttal and other reviews. The rebuttal addresses my questions, and my opinion on accepting the paper remains positive.

[Author Response · NeurIPS 2020]

We thank the reviewers for their constructive feedback and suggestions. We deeply appreciate their efforts in these difficult times. It is very encouraging that the reviewers found our dataset valuable (R3) and well-tailored (R2); our data collection strategy rational (R3) and founded (R1); construction of minimally contrastive pairs clever (R3); and our paper clear and well-written (R1, R2, R3).

**R2: This paper does not provide a proper solution to fully leverage this dataset to advance the relation reasoning from 2D.** Proposing a proper solution to relational reasoning in 2D is important but out of scope for this paper. We hope that our work can help catalyze such advances. Although our primary contribution is a dataset for grounding spatial relations in 3D, we have provided insights that can be useful in advancing relation reasoning from 2D. Our results suggest that estimating 3D configurations could be a promising first step for understanding spatial relations in 2D. Also, our dataset exposes that recent 2D methods overly rely on bias, and hence can help build better models that overcome these shortcomings.

**R3: Discuss how the work relates to earlier works by Chang et al.** Thanks for bringing these works to our notice. Indeed the earlier works by Chang et al. are related to ours. In [Chang et al. EMNLP 2014], the authors model spatial knowledge by leveraging statistics in 3D scenes. For spatial relations, they create a dataset with 609 annotations between 131 object pairs in 17 scenes. We on the other hand, collect a relatively larger dataset for spatial relation grounding with 10K annotations, across 5K object pairs in 5K scenes. In [Chang et al. ACL 2015], the authors focus on creating a model for generating 3D scenes from text, and create a dataset of 1129 scenes from 60 seed sentences. We also create 3D scenes, but unlike [Chang et al. ACL 2015] our dataset focuses on only spatial relations not other properties like object color. Moreover, unlike [Chang et al. EMNLP 2014, Chang et al. ACL 2015], scenes in Rel3D occur in minimally contrastive pairs which control for potential biases like language bias. Finally, the objects in [Chang et al. EMNLP 2014, Chang et al. ACL 2015] are limited to those found in indoor scenes like chairs and tables, while we also consider outdoor objects like trees, planes, cars and birds. Hence Rel3D covers a wider array of spatial relations. We will add this discussion to the paper.

**R3: How were specific object instances selected and were they pre-specified for a given crowdworker?** The specific object instances given to crowdworker were based on the expert annotator' response. In the first stage of data collection, we showed randomly-selected object instances to the expert annotators. Based on the object instances, the expert annotator marked if a particular relation between them is natural. When asking the crowdworker to create the scene, we gave them the pre-selected object instances. Also, the crowd-workers had an option to select "Not Possible", if they deemed that the relation could not hold between the object instances. However, this option was used very infrequently ($< 5\%$ of times).

**R3: Rationale for subsampling only 1/4 of all relations and what relations are excluded due to that?** This is due to the limited budget for expert annotators. There are 134,670 possible relations and it would be too costly to annotate each of them as natural or not.

**R3: To my understanding, a crowdworker's judgment that a relation is not valid in a presented image is deemed to imply that the relation is not using an intrinsic frame of reference – is this a reasonable assumption (i.e. are there other cases when a crowdworker might say the relation does not hold)?** The assumption is reasonable because the same worker is asked to judge the relation from a different view that would negate the relation unless the worker is using an intrinsic frame. While collecting a scene, we first ask a crowdworker to place the subject and object so that they satisfy the relation. Once done, we show the same worker an image taken from a different camera view. For "to the left of", "to the right of", "in front of" and "behind", the new camera view is directly opposite from the initial one, i.e. from the back wall. For the relation "to the side of"; the new camera view is from the left wall. Since the task and the image are shown in quick succession, we assume that the worker would use the same frame of reference. Hence, if the worker did the task correctly but says that the relation is not valid in a different view, it implies that they did not choose an intrinsic frame of reference (intrinsic to the object) while doing the task, but rather did the task from an extrinsic frame of reference that is dependent on the camera view.
A possible corner case could arise if one object is completely blocked by another in the second view. But this is largely avoided by using a slanted topish view (as shown in Figure 2). Lastly, in the final filtering step, we ask independent workers to verify the relation while taking into account the inferred frame of reference. Hence any erroneous sample is filtered out in the final dataset.

**R1: The scenes involve only two objects and thus not so realistic.** We agree that more objects can improve realism. With 2 objects per scene, our data is still useful because it provides a stepping stone for more different cases. In addition, Rel3D can be easily augmented by inserting distractor objects or by merging multiple scenes, without collecting additional human annotations.

[Meta-Review · NeurIPS 2020]

This submission proposes a new benchmark for visual understanding of spatial object relationships in 3D. It initially received three reviews with all positive scores (8,6,7). The reviewers appreciated the scale and the efforts put in making this dataset balanced, but also noted that it is not completely realistic (only two objects in each scene). The rebuttal addressed other concerns of the reviewers. For these reasons, the AC's recommendation is to accept this submission as a spotlight.